# Pessimistic Backward Policy for GFlowNets

**Hyosoon Jang[1], Yunhui Jang[1], Minsu Kim[2], Jinkyoo Park[2], Sungsoo Ahn[1]**
[1]POSTECH     [2]KAIST
{hsjang1205,uni5510,sungsoo.ahn}@postech.ac.kr,
{min-su,jinkyoo.park}@kaist.ac.kr

## Abstract

This paper studies Generative Flow Networks (GFlowNets), which learn to sample objects proportionally to a given reward function through the trajectory of state transitions. In this work, we observe that GFlowNets tend to under-exploit the high-reward objects due to training on insufficient number of trajectories, which may lead to a large gap between the estimated flow and the (known) reward value. In response to this challenge, we propose a pessimistic backward policy for GFlowNets (PBP-GFN), which maximizes the observed flow to align closely with the true reward for the object. We extensively evaluate PBP-GFN across eight benchmarks, including hyper-grid environment, bag generation, structured set generation, molecular generation, and four RNA sequence generation tasks. In particular, PBP-GFN enhances the discovery of high-reward objects, maintains the diversity of the objects, and consistently outperforms existing methods.

## 1 Introduction

Generative Flow Networks [1, GFlowNets] are models that sample compositional objects from a Boltzmann distribution defined by some reward function. To this end, GFlowNets construct an object through a trajectory of state transitions, e.g., iteratively adding molecular fragments to construct a molecule. They are attractive for their ability to sample a diverse set of high-reward objects, as demonstrated in molecular discovery [2, 3], biological sequence design [4], combinatorial optimization [5], and large language models [6].

In detail, GFlowNets aim to sample from the Boltzmann distribution using a *forward policy* to decide the state transitions. However, this is challenging since the forward policy induces the distribution over trajectories, while the Boltzmann distribution is only defined on the terminal state of trajectories, i.e., objects. Hence, directly matching the two distributions with respect to the terminal state requires an intractable marginalization of the forward policy over the exponentially sized trajectory space.

To circumvent this issue, GFlowNets employ an auxiliary *backward policy* that lifts the Boltzmann distribution to the trajectories via reversing the state transitions. In particular, the backward policy decomposes the unnormalized Boltzmann density of a terminal state into the unnormalized densities of trajectories, coined *backward flow*, associated with the terminal state. Then the forward policy learns the Boltzmann distribution by matching its unnormalized density, i.e., reward, coined *forward flow*, with the backward flow on the observed trajectories. We call this training scheme *flow matching*.[1]

The training objective of the flow matching has been investigated such as detailed balance [7] and trajectory balance [8], and sub-trajectory balance [9]. To facilitate training, improved credit assignment techniques have been explored [10, 11]. Additionally, exploration methods [12] have been proposed to collect more diverse trajectories. Moreover, exploitation methods such as focusing on the higher-reward trajectories from the backward policy [13] and sampling high-reward trajectories with local search [14] have been presented for the collection of higher-reward trajectories.

---

[1]In this work, we refer to flow matching as the learning scheme that aligns the forward and backward flow.

38th Conference on Neural Information Processing Systems (NeurIPS 2024).

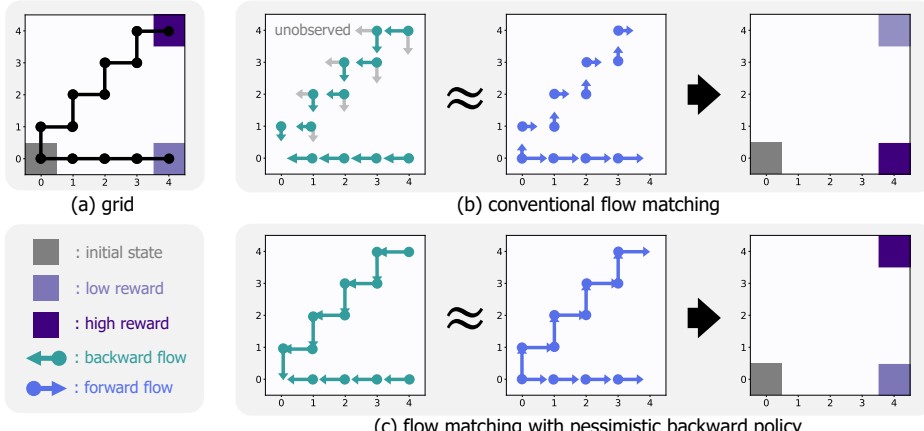

Figure 1: **Flow matching for observed trajectories. (a)** The task aims to reach the terminal state with a reward-proportional probability from the initial state, by incrementing one coordinate as a random action. The black line indicates the two observed trajectories for each terminal state. **(b-c)** The arrow ($\rightarrow$) length indicates the amount of the backward or forward flow. In **(b)**, the flow matching ($\approx$) between the observed backward and forward flows underestimates the high-reward object due to the low observed backward flow. In **(c)**, PBP-GFN succeeds with the observed backward flow that fully represents the true rewards.

In this work, we point out a pitfall of the flow matching objectives: when only a small portion of object-sharing trajectories is observed, GFlowNets tend to under-exploit the object. This pitfall stems from the under-determination of the forward flow, due to training only on the observed backward flow that partially represents the true high reward. Consequently, the forward policy tends to assign high probabilities to objects with high observed backward flow, rather than the high reward objects, as illustrated in (a) and (b) of Figure 1. This is counter-intuitive as the forward policy favors objects with low rewards despite possessing the knowledge of other objects with higher rewards. While one could bypass this issue at the cost of observing more trajectories [13], we pursue an alternative direction in this work.

We propose a simple remedy for the under-exploitation problem: a pessimistic backward policy for GFlowNets (PBP-GFN). Our key idea is the maximization of the observed backward flow to align the observed backward flow to the true reward. Consequently, PBP-GFN resolves the under-exploitation problem which favors the object with high observed backward flow while neglecting the true reward, as illustrated in (c) of Figure 1. We also note that our algorithm preserves the asymptotic optimality to induce the target Boltzmann distribution by simply modifying the backward policy while preserving the true rewards [8]. Additionally, we analyze how our algorithm reduces the error bound in estimating the true Boltzmann distribution.

We extensively validate PBP-GFN on various benchmarks: hyper-grid benchmark [1], bag generation [13], maximum independent set problem [5], fragment-based molecule generation [1], and four RNA sequence generation tasks [4]. In these experiments, we observe that PBP-GFN (1) improves the learning of target Boltzmann distribution and (2) enhances the discovery of high-reward objects, while (3) maintaining the diversity of the sampled high-reward objects.

To conclude, our contributions can be summarized as follows:

- We characterize the under-exploitation problem stemming from an under-determined flow that only learns the observed flow for partially observed trajectories (Example 1).
- To resolve this issue, we propose pessimistic training of backward policy that aims to reduce the amount of unobserved flow for the observed objects.
- Through extensive experiments, we show that our algorithm consistently improves the performance of GFlowNets compared to prior works for designing the backward policy, even higher than other training algorithms for discovering high-reward objects.[2]

---

[2]Code: https://github.com/hsjang0/Pessimistic-Backward-Policy-for-GFlowNets.

## 2 Preliminaries

Generative Flow Networks [1, 7, GFlowNets] generate an object $x$ from the object space $\mathcal{X}$ through a trajectory $\tau = (s_0, s_1, \ldots, s_T)$ of state transitions, where the terminal state is the object $s_T = x \in \mathcal{X}$ to be generated. Here, a forward policy $P_F(s_{t+1}|s_t)$ makes the transition from the state $s_t$ to the next state $s_{t+1}$ and assigns a probability of $P_F(\tau) = \prod_{t=0}^{T-1} P_F(s_{t+1}|s_t)$ to the trajectory $\tau$.

Next, GFlowNets train the forward policy to sample objects from a Boltzmann distribution defined by a reward function $R(x)$ that satisfies:

$$P_F^\top(x) \propto R(x). \tag{1}$$

Here, $P_F^\top(x)$ is a distribution of an object $x$ marginalized over exponentially sized non-terminal state spaces. To circumvent this intractability, GFlowNets train on the flow matching objectives.

**Flow matching for training GFlowNets.** To learn the Boltzmann distribution, the forward policy $P_F$ aims to align to a backward policy $P_B$. The backward policy $P_B(\tau|x) = \prod_{t=0}^{T-1} P_B(s_t|s_{t+1})$ decomposes the reward into the unnormalized densities of object-sharing trajectories $\mathcal{T}(x)$ for an object $x$, i.e., $R(x) = \sum_{\tau \in \mathcal{T}(x)} R(x)P_B(\tau|x)$.

To be specific, the forward policy learns to match the unnormalized densities to the backward policy, coined *flow matching*, over all trajectories in the trajectory space $\mathcal{T}$:

$$\forall \tau \in \mathcal{T} \quad Z_\theta P_F(\tau) \approx R(x)P_B(\tau|x), \tag{2}$$

where $Z_\theta P_F(\tau)$ is a *forward flow* defined with a learnable constant $Z_\theta$, and $R(x)P_B(\tau|x)$ is a *backward flow*. Equation (2) induces the forward policy following Boltzmann distribution, i.e., $Z_\theta P_F^\top(x) \approx R(x)$, by marginalizing trajectory flows over set of trajectories $\mathcal{T}(x)$ inducing the object $x$, i.e., $\sum_{\tau \in \mathcal{T}(x)} Z_\theta P_F(\tau) \approx Z_\theta P_F^\top(x)$ and $\sum_{\tau \in \mathcal{T}(x)} R(x)P_B(\tau|x) \approx R(x)$.

To satisfy Equation (2), GFlowNets minimize various training objectives. One such objective is the trajectory balance [8, TB], defined as follows:

$$\mathcal{L}_{\text{TB}}(\tau) = \left( \log \frac{Z_\theta P_F(\tau)}{R(x)P_B(\tau|x)} \right)^2, \tag{3}$$

which is minimized over trajectories observed during training, e.g., trajectories sampled from the forward policy. The set of observed trajectories stored in the replay buffer inducing the object $x$ is denoted as $\mathcal{B}(x) \subset \mathcal{T}(x)$. Note that training objectives for Equation (2) can also be defined on a transition [7, DB] or sub-trajectories [8, subTB].

## 3 Method

In this section, we introduce our pessimistic backward policy for generative flow networks (PBP-GFN). First, we show that forward policies trained with flow matching tend to under-exploit high-reward object $x$ with partially observed trajectories $\mathcal{B}(x)$ when the underdetermined forward flow only learns the small amount of observed backward flow for the high-reward object (Section 3.1). To address this issue, we propose pessimistic training of backward policy that increases the proportion of observed flow for the object, which leads to an accurate estimation of the reward (Section 3.2).

### 3.1 Motivation: under-exploitation of objects with partially observed trajectorie

First, we explain how conventional flow matching may suffer from the under-exploitation of observed high-reward objects. To this end, we decompose the reward $R(x)$ into two components: (1) *observed backward flow* $R_{\mathcal{B}}(x) = \sum_{\tau \in \mathcal{B}(x)} R(x)P_B(\tau|x)$ assigned to the partially observed trajectories $\mathcal{B}(x)$, and (2) *unobserved backward flow* $R(x) - R_{\mathcal{B}}(x)$ assigned to the unobserved trajectories $\mathcal{T}(x) \backslash \mathcal{B}(x)$. Then (1) and (2) are paired with *observed forward flow* and *unobserved forward flow*, respectively.

In detail, on the one hand, conventional flow matching aligns the observed forward flow to the observed backward flow for the first component, i.e., $\sum_{\tau \in \mathcal{B}(x)} Z_\theta P_F(\tau) \approx R_{\mathcal{B}}(x)$. On the other hand, there exists degree of freedom for the unobserved forward flow $\sum_{\tau \in \mathcal{T}(x) \backslash \mathcal{B}(x)} Z_\theta P_F(\tau)$, as it is challenging to match the flow over unobserved trajectories, i.e. trajectories not in the buffer $\mathcal{B}$.

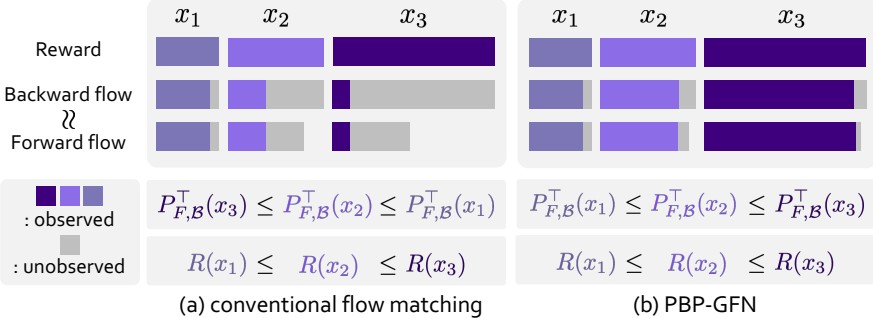

Figure 2: **Under-exploitation of objects with partially observed trajectories.** The reward $R(x)$ consists of (1) observed backward flow $R_{\mathcal{B}}(x)$ and (2) unobserved backward flow $R(x) - R_{\mathcal{B}}(x)$. **(a)** Conventional flow matching may assign a higher probability to the lower-reward object as the observed forward flow is aligned only with a small amount of observed backward flow. This fails to assign the accurate probability proportional to the reward. **(b)** PBP-GFN assigns more accurate probability proportional to the reward, by increasing the proportion of observed flow.

Overall, flow matching induces a forward policy with the marginalized probability $P_{F,\mathcal{B}}^{\top}(x)$:

$$P_{F,\mathcal{B}}^{\top}(x) \propto \left( R_{\mathcal{B}}(x) + \sum_{\tau \in \mathcal{T}(x) \setminus \mathcal{B}(x)} Z_\theta P_F(\tau) \right). \tag{4}$$

Here, our key observation is that, for an observed object $x$ with a *high reward* $R(x)$ but *a small amount of observed backward flow* $R_{\mathcal{B}}(x)$, the unobserved backward flow $R(x) - R_{\mathcal{B}}(x)$ is likely to be much larger than the forward flow of the unobserved trajectories $\sum_{\tau \in \mathcal{T}(x) \setminus \mathcal{B}(x)} Z_\theta P_F(\tau)$. This leads to the under-exploitation of the high-reward object $x$, since the forward policy assigns a higher probability to another object $x'$ with a *lower reward* but *a larger amount of observed flow* $R_{\mathcal{B}}(x)$, as illustrated in (a) of Figure 2. As a result, the marginalized probability $P_{F,\mathcal{B}}^{\top}(x)$ may converge to a local optimum that yields a smaller expected reward compared to the target Boltzmann distribution.

To further motivate our proposal regarding the under-exploitation problem, we present a failure case of flow matching converged to a local optimum contradicting the observed rewards in Example 1. We construct a particular instance of Equation (4) where the forward policy underestimates the high-reward object compared to the lower one. We depict this example in (a) and (b) of Figure 3.

**Example 1.** *Consider two objects $x_1$ and $x_2$ with rewards of $1$ and $\frac{1}{2}$, respectively, where $x_1$ is reached by three trajectories ($|\mathcal{T}(x_1)| = 3$) and $x_2$ is reached by one ($|\mathcal{T}(x_2)| = 1$). Here, one trajectory for each object is observed ($|\mathcal{B}(x_1)| = |\mathcal{B}(x_2)| = 1$). Then, the probability to induce object $x_1$ can be assigned as $P_{F,\mathcal{B}}^{\top}(x_1) \propto \frac{1}{3}$ since the forward flow still matches the backward flows for the observed trajectory $\tau_1$, i.e., $P_{F,\mathcal{B}}^{\top}(x_1) = P_{F,\mathcal{B}}(\tau_1) \propto R(x_1)P_B(\tau_1|x_1) = \frac{1}{3}$. This is lower than $P_{F,\mathcal{B}}^{\top}(x_2) \propto \frac{1}{2}$ assigned with fully observed trajectories.*

Example 1 is counter-intuitive, as a higher probability is assigned to the lower reward object $x_2$ despite observing the higher reward object $x_1$. The forward policy $P_F(\tau)$ also assigns zero probability to unobserved trajectories. Consequently, the probability $P_{F,\mathcal{B}}^{\top}(x)$ cannot be corrected even more trajectories are sampled from policy $\tau \sim P_{F,\mathcal{B}}(\tau)$. This hints at the necessity of the remedy for the under-exploitation of objects due to the small amount of observed flow, with the fixed observation $\mathcal{B}$.

## 3.2 Pessimistic backward policy for GFlowNets

Here, we propose a pessimistic training method for the backward policy in GFlowNets, coined PBP-GFN, which aims to resolve the under-exploitation problem of the flow matching with partially observed trajectories introduced in Section 3.1. To address this challenge, the backward policy is trained to reduce the amount of unobserved backward flow, being pessimistic about unobserved trajectories inducing the observed object. It is notable that the total backward flow for the object, i.e., reward, is preserved by shifting the unobserved backward flow into the observed backward flow.

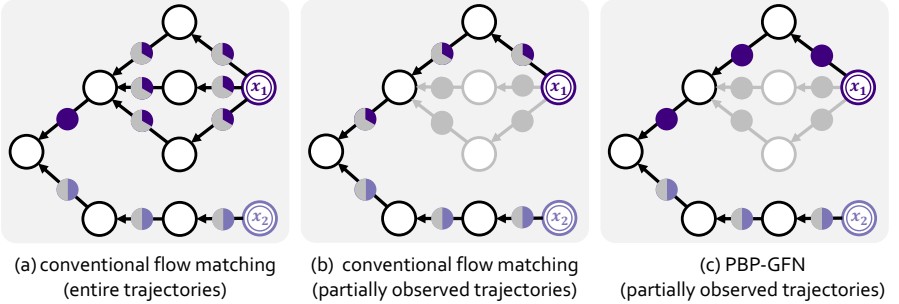

| (a) conventional flow matching (entire trajectories) | (b) conventional flow matching (partially observed trajectories) | (c) PBP-GFN (partially observed trajectories) |

Figure 3: **Pessimistic backward policy for GFlowNets (PBP-GFN).** The portion of the circle indicates the amount of flow, e.g., ● indicates the flow of 1, and ◑ indicates the half flow of ●, i.e., the flow of 0.5. Additionally, the color of the flow indicates the flow inducing the same-colored reward, and the black and gray lines indicate the observed and unobserved trajectories, respectively. (**a**) Flow matching succeeds with the entire trajectories. One can observe that the true reward of $x_1$ is 1 and the reward of $x_2$ is 0.5 by the amount of flow. (**b**) Flow matching fails with partially observed trajectories. (**c**) PBP-GFN assigns high probabilities to the backward transitions of observed trajectories to keep a high probability to high-reward objects.

---

**Algorithm 1** Learning pessimistic backward policy for GFlowNets

---

1: Initialize the replay buffer (i.e., the set of observed trajectories) $\mathcal{B}$, forward policy $P_F$, backward policy $P_B$, and parameter $Z_\theta$.
2: **repeat**
3:     Sample a batch of trajectories $\{\tau^{(k)}\}_{k=1}^{K}$ from the behaviour policy.
4:     Update $\mathcal{B} \leftarrow \mathcal{B} \cup \{\tau^{(k)}\}_{k=1}^{K}$.
5:     **for** $n = 1, \ldots, N$ **do**                    ▷ *Learning pessimistic backward policy*
6:         Update $P_B$ to minimize $\ell_{\text{PBP}}$ over $\mathcal{B}$ with stochastic gradients.
7:     **end for**
8:     Update $P_F, Z_\theta$ to minimize $\mathcal{L}_{\text{TB}}$ with $\{\tau^{(b)}\}_{k=1}^{K}$.
9: **until** converged

---

To be specific, given a replay buffer $\mathcal{B}$, the pessimistic training of backward policy $P_B$ aims to increase the backward flow for the observed trajectories ending with the object $x$. Specifically, it aims to maximize $R_\mathcal{B}(x) = \sum_{\tau \in \mathcal{B}(x)} R(x) P_B(\tau|x)$, thereby aligning the observed backward flow $R_\mathcal{B}(x)$ to the true reward $R(x)$, reaching the upper bound $R_\mathcal{B}(x) \approx R(x)$. Consequently, flow matching with such a backward flow for partially observed trajectories induces an observed forward flow that accurately estimates the true reward, thereby preventing the under-exploitation of the rewards due to the small amount of observed flow (Example 1), as illustrated in (b) of Figure 2 and (c) of Figure 3.

Furthermore, it is worth noting how PBP-GFN better estimates the Boltzmann density, i.e., reward. The high-level idea is that, given the fixed total flow, maximizing the observed forward and backward flows with PBP-GFN naturally minimizes the unobserved forward and backward flows, thereby reducing a flow matching error for the unobserved flows effectively. The detailed error bound in estimating the Boltzmann density is described in Appendix A.

**Pessimistic training of backward policy.** We train the parameterized backward policy $P_B$ to increase the backward trajectory flows in observed trajectories, $R(x) \sum_{\tau \in \mathcal{B}_x} P_B(\tau|x)$, by assigning higher probabilities to the backward transitions $P_B(\tau|x) = \prod_{t=0}^{T-1} P_\mathcal{B}(s_t|s_{t+1})$ of observed trajectories, i.e., $\sum_{\tau \in B(x)} P_B(\tau|x) \approx 1$. We achieve this by minimizing the negative log-likelihood:

$$\ell_{\text{PBP}} = -\mathbb{E}_{\tau \in \mathcal{B}(x)}[\log P_B(\tau|x)], \tag{5}$$

where $x$ is the object induced by the trajectory $\tau$. It is notable that our approach only modifies the relative backward trajectory flows among trajectories inducing the same object and does not alter the total amount of backward flows, thereby preserving the asymptotic optimality of flow matching for learning the target Boltzmann distribution. Note that the training of the pessimistic backward policy is practical in most cases, as it only requires computing the stochastic gradients to minimize $\ell_{\text{PBP}}$.

Subsequently, we train the GFlowNets with the learned pessimistic backward policy. The pessimistic backward policy is learned online with the forward policy, as new trajectories are observed for the training in each round. The training algorithm is described in Algorithm 1.[3] Note that the pessimistic training is agnostic to the choice of flow matching objectives [7–9].

# 4 Related work

**Generative Flow Networks (GFlowNets).** GFlowNets [1, 7] train a forward policy that sequentially constructs objects sampled from a Boltzmann distribution. They are closely related to reinforcement learning in soft Markov Decision Processes (soft MDPs) [15–17] and variational inference [18]. Recently, there has been a surge in research on improving the training of GFlowNets, such as introducing novel flow matching objective functions [8, 9, 12, 13], enhancing off-policy exploration [14, 19–21], incorporating order information for enabling preference-based optimization [22], and improving credit assignment [10, 11]. Moreover, GFlowNets are increasingly applied across a wide range of fields such as molecular optimization [2, 3, 23], biological sequence design [4, 24], probabilistic modeling and inference [25, 26], combinatorial optimization [5, 27, 28], continuous stochastic control [29–31], and large language models [6].

Despite the advancements in training GFlowNets, there still exists the challenge of dealing with the vast number of trajectories. The number of trajectories grows exponentially with the increase in the number of state spaces and actions, making it impractical to observe all trajectories during training. This issue can be partially addressed by facilitating the discovery of unobserved trajectories [12]. However, the problem of probability not matching the rewards remains unless sampled trajectories comprehensively cover all possible flows.

**Training GFlowNets with auxiliary backward policy.** GFlowNets train a forward policy to align with the auxiliary backward policy, which inverts the construction process of the object. Therefore, the choice of the backward policy directly impacts the training of GFlowNets and is vital to the improvement of the sampling performance. Despite its crucial role, the choice of backward policy has gained limited attention with only a few works [8, 13, 15], and none of these works tackle the under-exploitation of high-reward objects caused by unobserved backward flow.

For instance, while the uniform [8] and the MaxEnt [15] backward policies assign a fixed probability to the backward transition for enhancing exploration, our pessimistic backward policy learns the backward transition probability for enhancing exploitation. Next, conventional [8] and sub-structure [13] backward policies may enhance the exploitation by learning the backward flow to align with the forward flow or to improve the credit assignments. However, they do not directly reduce the unobserved backward flow and do not resolve the under-exploitation stemming from that.

# 5 Experiment

We evaluate our method on various domains, including a hyper-grid [1], bags [13], structured sets [5], molecules [1], and RNA sequences [13, 14]. As base metrics, we consider the number of modes, e.g., samples with rewards higher than a specific threshold, and the average top-100 score, which are measured via samples collected during training. We report the performances using three different random seeds. In these experiments, one can observe that:

- PBP-GFN improves learning of the target Boltzmann distribution (Figures 4 and 6).
- PBP-GFN enhances the discovery of high-reward objects (Figures 5, 7 and 8).
- PBP-GFN maintains the diversity of sampled high-reward objects and promotes the discovery of distinct diverse modes (Figures 7(c) and 9).

## 5.1 Synthetic tasks

In synthetic environments, i.e., hyper-grid environment, bag generation, and maximum independent set, we first show how our method (PBP-GFN) improves the performance compared to the prior methods that proposed various designs of the backward policy, on both the trajectory balance [8, TB] and detailed balance-based implementations [7, DB]. As baselines, we consider the conventional

---

[3]We describe the detailed implementations in Appendix B.

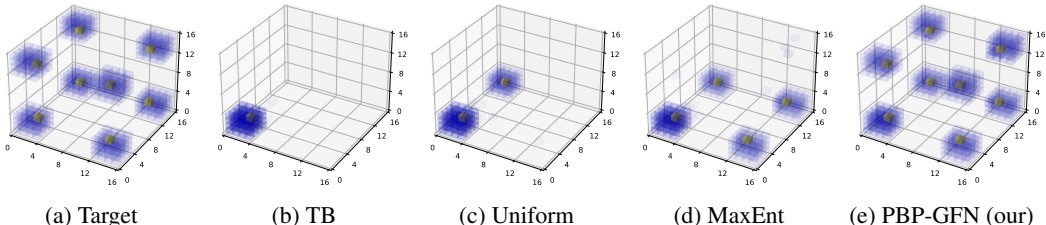

| (a) Target | (b) TB | (c) Uniform | (d) MaxEnt | (e) PBP-GFN (our) |

Figure 4: **The target distribution and empirical distributions of each model trained with $10^5$ trajectories**. The empirical distributions are computed as rescaled products of the distribution over three runs. Our method (PBP-GFN) consistently discovers all modes over three runs and learns the target Boltzmann distribution correctly within the relatively small number of trajectories.

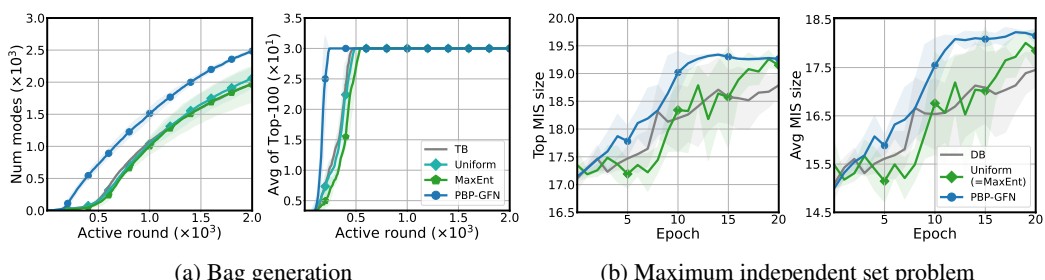

| (a) Bag generation | (b) Maximum independent set problem |

Figure 5: **The performance comparison with the prior backward policy design methods.** The solid line and shaded region represent the mean and standard deviation, respectively. The PBP-GFN shows superiority in generating diverse high-reward objects, compared to the considered baselines for designing the backward policy.

backward policy trained with the TB or DB [7], the uniform backward policy [8], and the maximum entropy backward policy [15, MaxEnt].

**Hyper-grid [1].** We first consider the hyper-grid, where the target Boltzmann distribution is defined over the $16 \times 16 \times 16$ grid illustrated in Figure 4(a). We also consider the $20 \times 20 \times 20 \times 20$ hyper-grid. The actions are incrementing one coordinate by one or terminating. The high-reward regions, i.e., modes, are defined as near the corners of the grid that are separated by regions with very small rewards. In this task, we consider the TB-based implementation following the prior work [8]. The detailed experimental settings are described in Appendix B. In this task, we measure the L1 distance between the target Boltzmann distribution and the empirical distribution of $P_F^\top(x)$, with the measurable likelihood of the Boltzmann distribution.

**Bag generation [13].** We next consider a simple bag generation task, where the action is adding an item to a bag. The bag yields a high reward when seven repeated items are included, i.e., modes. We apply our method to the prior TB-based implementation on this task [13] and compare it with TB and MaxEnt. The detailed setting is described in Appendix B.

**Maximum independent set [5].** We also consider solving maximum independent set problems, where the action is selecting a node and the reward is the size of the independent set. At each epoch, the GFlowNets train with the set of training graphs, and sample 20 solutions for each validation graph and measure the average reward and the maximum reward following Zhang et al. [5]. We apply our method to the prior DB-based implementation of this task [5]. The experimental setting is described in Appendix B. Note that the MaxEnt is equivalent to the uniform backward policy in this task.

**Results.** In Figure 4 and Figure 6, we depict the empirical sampling distribution and the L1 distance from the target Boltzmann distribution for each method in the hyper-grid environment. Here, one can see that our method (PBP-GFN) captures all modes and converges to the target Boltzmann distribution faster than the baselines. These results can be attributed to the capabilities of PBP-GFN, which enables us to effectively learn from the large amount of correct forward flow.

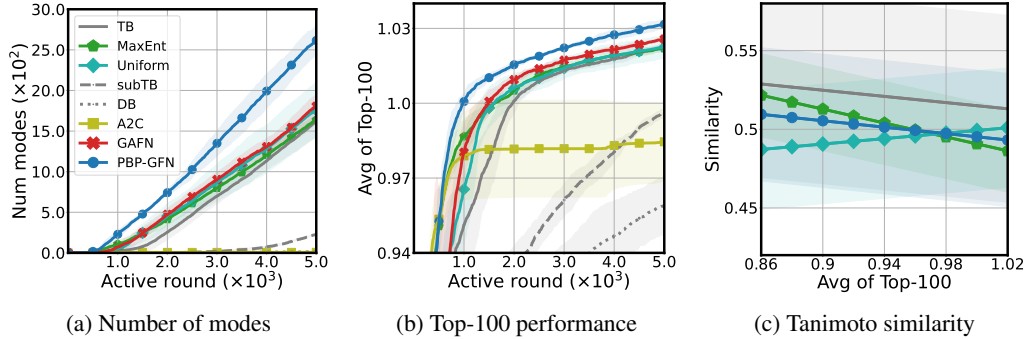

| (a) Number of modes | (b) Top-100 performance | (c) Tanimoto similarity |
|---|---|---|

Figure 7: **The performance on molecular generation.** The solid line and shaded region represent the mean and standard deviation, respectively. The PBP-GFN shows superiority compared to the baselines in generating diverse high reward molecules while yielding similar Tanimoto similarities compared to other baselines with prior backward policy designs.

In both bag generation and maximum independent set problems, one can see that our approach also shows (1) superior performance (2) or faster convergence compared to the baselines as illustrated in Figure 5. One can reason this result stems from the capabilities of PBP-GFN that facilitate the learning of correct Boltzmann distribution. Furthermore, it is worth noting that our method makes improvements over both TB and DB-based implementations.

## 5.2 Molecular generation

Next, we evaluate our method in the fragment-based molecule generation [1], where the action is adding a molecular building block. The reward is the binding energy between the molecule and the target protein computed by a pre-trained oracle [1]. Here, the mode is defined as a high-reward molecule with a low Tanimoto similarity [32] measured against previously accepted modes.

We consider a TB-based implementation for our method and compare with various baselines including GFlowNets and reinforcement learning algorithms: DB, sub trajectory balance [9, subTB], TB, TB defined with uniform and MaxEnt backward policies, generative augmented flow networks [10, GAFN], and Advantage Actor-Critic [33, A2C]. For the evaluation metric, we analyze the trade-off between the average score of the top 100 samples and the diversity of these samples. Additionally, to measure diversity, we compute the average pairwise Tanimoto similarity following prior works. The detailed setting is described in Appendix B.

**Results.** We depict the results in Figure 7. One can see that our method, i.e., PBP-GFN, outperforms the baselines in enhancing the average score of unique top-100 molecules and the number of

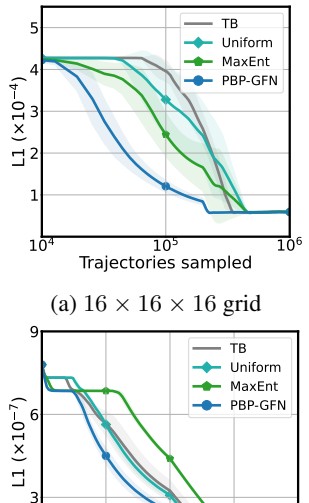

(a) $16 \times 16 \times 16$ grid

(b) $20 \times 20 \times 20 \times 20$ grid

Figure 6: **L1 distance between Boltzmann distribution.** PBP-GFN shows fastest learning target distribution with respect to the observed trajectories.

modes found during training. These results highlight that PBP-GFN also can make improvements for environments with a huge state space. Furthermore, one can see that our method yields low Tanimoto similarities between top-100 molecules with respect to the average reward. This verifies that our algorithm not only generates high-scoring samples but also diverse molecules.

## 5.3 Sequence generation

We consider four RNA sequence generation tasks that aim to discover diverse and promising sequences that bind to human transcription factors [4, 34, 35], where the action is appending or prepending an amino acid. As baselines, we consider the same baselines as in the fragment-based molecule generation. In this task, we conduct experiments on the following four benchmarks.

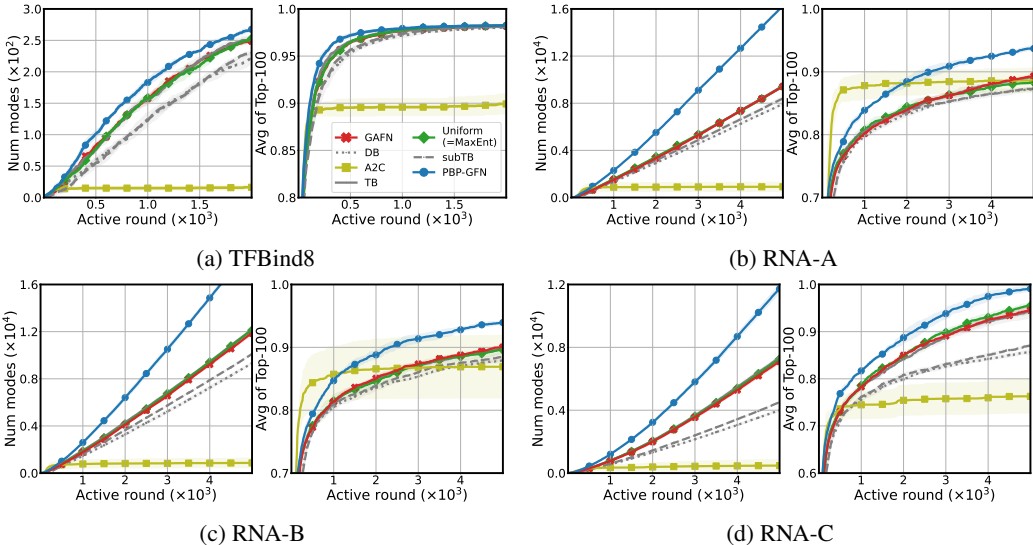

Figure 8: **The performance on RNA sequence generation.** The solid line and shaded region represent the mean and standard deviation, respectively. The PBP-GFN shows superiority compared to the baselines in generating diverse high reward sequences.

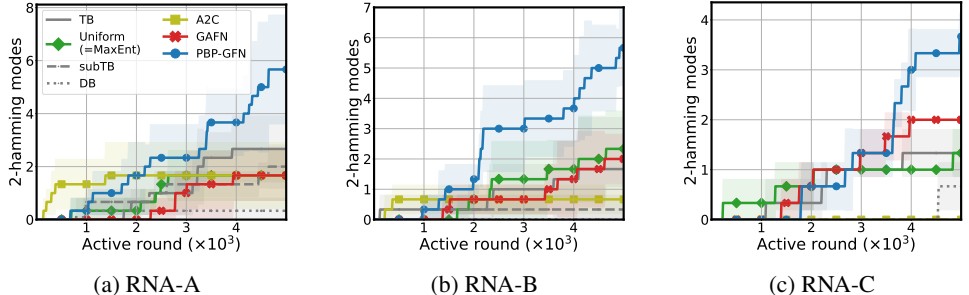

Figure 9: **The number of 2-hamming ball modes discovered during training.** The solid line and shaded region represent the mean and standard deviation, respectively. The PBP-GFN shows superiority compared to the baselines in discovering diverse distinct modes.

**TFBind8.** This is the task to generate length-eight RNA sequences. The reward is computed by wet-lab measured DNA binding activity to Sine Oculis Homeobox Homolog 6 [34]. The mode is determined based on whether it is included in a predefined set of promising RNA sequences [13].

**RNA-Binding.** This task generates length-14 RNA sequences. In this task, we consider three different target transcriptions: RNA-A, RNA-B, and RNA-C [14, 36]. The mode is defined as an RNA sequence with a reward higher than the threshold. In this task, we also consider the 2-hamming ball modes [36], which is defined as the local maximum among its intermediate neighborhoods defined by modifying $n$ components of the sequence.

**Results.** The results are presented in Figure 8. One can see that FBP-GFN shows faster convergence or superior performance compared to the considered baselines in enhancing the average score of unique top-100 sequences and the number of modes during training. Furthermore, in Figure 9, one can see that our method better discovers the diverse distinct modes that are separated far from each other, compared to the baselines.

## 5.4 Ablation studies

**Comparing overall generated sample quality.** To further analyze the overall sample quality, we provide the relative mean error [13] which measures the distance between the mean values of the empirical generative distribution and the target Boltzmann distribution. We present the results in Figure 10. One can see that our method yields the lowest errors.

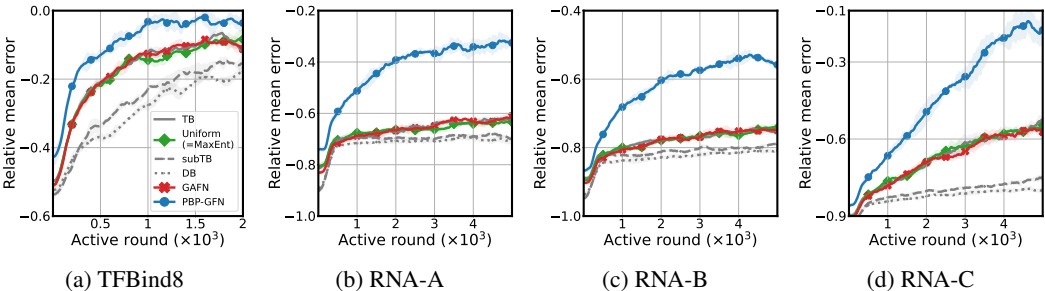

(a) TFBind8  (b) RNA-A  (c) RNA-B  (d) RNA-C

Figure 10: **The relative mean error comparison.** The solid line and shaded region represent the mean and standard deviation, respectively. Our PBP-GFN yields the closest error to zero.

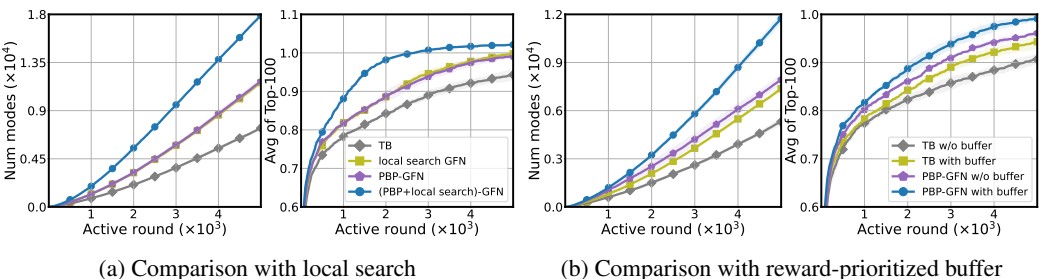

(a) Comparison with local search  (b) Comparison with reward-prioritized buffer

Figure 11: **Comparison with off-policy sampling methods for exploitation.** The considered task is RNA-C. The solid line and shaded region represent the mean and standard deviation, respectively. One can see that PBP-GFN (1) shows competitive performance compared to GFNs with local search and reward-prioritized buffers, and (2) improves performance when combined with them.

**Comparison with other exploitation methods.** We further validate PBP-GFN by comparing or combining the pessimistic backward policy with local search [14] and reward-prioritized buffer [13] that are off-policy sampling methods and orthogonal to our pessimistic backward policy. We present the experimental results in Figure 11. One can observe that our method (1) shows similar performance compared to them and (2) consistently improves the performance when combined with them.

## 6 Conclusion

In this work, we identify the under-exploitation problem in flow matching due to the large amount of unobserved flow lifted by the backward policy. To resolve this, we introduce a pessimistic training of the backward policy for GFlowNets (PBP-GFN) that reduces the probabilities for unobserved backward trajectories leading to observed trajectories. Our PBP-GFN shows a successful alternative to the prior backward policies and has demonstrated improved performance across eight benchmarks.

**Limitation.** As an exploitation method, our PBP-GFN makes a trade-off between obtaining high-reward trajectories and diversified trajectories, i.e., there is no free lunch in the exploitation-exploration trade-off. Although our approach maintains the diversity of high-reward sampled objects in the considered benchmarks, this may not hold for some environments where exploration is significant. We discuss such a setting in Appendix C. To relax this issue, one can reduce the learning rate for the pessimistic backward policy or incorporate an exploration-focused off-policy sampling method. One can further control the trade-off by interpolating PBP-GFN with explorative GFNs, e.g., MaxEnt, which can be an interesting future work direction.

## Acknowledgements

This work partly was supported by Institute of Information & communications Technology Planning & Evaluation (IITP) grant funded by the Korea government(MSIT) (No. IITP-2019-0-01906, Artificial Intelligence Graduate School Program(POSTECH)), the National Research Foundation of Korea(NRF) grant funded by the Korea government(MSIT) (No. 2022R1C1C1013366), Basic Science Research Program through the National Research Foundation of Korea(NRF) funded by the

Ministry of Education(MSIT) (No. 2022R1A6A1A03052954), and GRDC(Global Research Development Center) Cooperative Hub Program through the National Research Foundation of Korea(NRF) funded by the Ministry of Science and ICT(MSIT) (No. RS-2024-00436165).

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

# A  Proof for error bound

We demonstrate that PBP-GFN yields a lower error bound between the true marginalized distribution $P_F^\top(x)$ and the target Boltzmann distribution $P_B^\top(x) = \frac{R(x)}{Z}$ defined with the $Z = \sum_{x \in \mathcal{X}} R(x)$, compared to the conventional GFlowNets. To this end, we derive the following bound:

$$\sum_x \left| P_F^\top(x) - P_B^\top(x) \right| \leq 2 - 2 \sum_{\tau \in \mathcal{B}(x)} P_B(\tau) + \epsilon. \tag{6}$$

where the $\epsilon$ depends on the error in the likelihood for trajectories over the observed trajectories, $\sum_{\tau \in \mathcal{B}(x)} (P_F(\tau) - P_B(\tau))$. In this equation, our pessimistic backward policy maximizes $\sum_{\tau \in \mathcal{B}(x)} P_B(\tau)$ by maximizing the likelihood of $\sum_{\tau \in \mathcal{B}(x)} P_B(\tau|x)$. This better reduces the error bound compared to the conventional backward policy.

The error bound in Equation (6) can be derived as follows:

$$\sum_x \left| P_F^\top(x) - P_B^\top(x) \right| = \sum_x \left| \sum_{\tau \in \mathcal{T}(x)} P_F(\tau) - \sum_{\tau \in \mathcal{T}(x)} P_B(\tau) \right|$$

$$\leq \sum_x \left| \sum_{\tau \in \mathcal{B}(x)} P_F(\tau) - \sum_{\tau \in \mathcal{B}(x)} P_B(\tau) \right| + \sum_x \left| \left( P_F^\top(x) - \sum_{\tau \in \mathcal{B}(x)} P_F(\tau) \right) - \left( P_B^\top(x) - \sum_{\tau \in \mathcal{B}(x)} P_B(\tau) \right) \right|$$

$$\leq \sum_x \left| \sum_{\tau \in \mathcal{B}(x)} P_F(\tau) - \sum_{\tau \in \mathcal{B}(x)} P_B(\tau) \right| + \sum_x \left| \left( P_F^\top(x) - \sum_{\tau \in \mathcal{B}(x)} P_F(\tau) \right) \right| + \sum_x \left| \left( P_B^\top(x) - \sum_{\tau \in \mathcal{B}(x)} P_B(\tau) \right) \right|$$

$$= \sum_x \left| \sum_{\tau \in \mathcal{B}(x)} P_F(\tau) - \sum_{\tau \in \mathcal{B}(x)} P_B(\tau) \right| + \sum_x \left( P_F^\top(x) - \sum_{\tau \in \mathcal{B}(x)} P_F(\tau) \right) + \sum_x \left( P_B^\top(x) - \sum_{\tau \in \mathcal{B}(x)} P_B(\tau) \right)$$

$$= \sum_x \left| \sum_{\tau \in \mathcal{B}(x)} P_F(\tau) - \sum_{\tau \in \mathcal{B}(x)} P_B(\tau) \right| + 2 - \sum_x \left( \sum_{\tau \in \mathcal{B}(x)} P_F(\tau) + \sum_{\tau \in \mathcal{B}(x)} P_B(\tau) \right)$$

$$= \epsilon + 2 - 2 \sum_{\tau \in \mathcal{B}(x)} P_B(\tau)$$

where the error $\epsilon$ is associated with the errors in trajectory flow matching over the observed trajectories $\sum_{\tau \in \mathcal{B}(x)} (P_F(\tau) - P_B(\tau))$.

# B  Experimental details

In all experiments, the backward policy is designed to have the same architecture as the forward policy, e.g., a feedforward network with the same hidden dimensions, but does not share the parameters with the forward policy. For all experiments, we set the learning rate for the pessimistic training of the backward policy as $1e-3$. Our overall implementations for each benchmark follow the prior studies. Note that we consider both on-policy and off-policy settings. We use a single GPU of NVIDIA GeForce RTX 3090.

**Hyper-grid environment.** The implementations follow the prior study by Malkin et al [8]. We consider $16 \times 16 \times 16$ hyper-grid, where the starting state is $(0, 0, 0)$ and the action is incrementing one coordinate or terminating. The reward is computed as follows:

$$R(s) = R_0 + 0.5 \prod_{d=1}^{D} \mathbb{I}\left[\left|\frac{s_d}{H-1} - 0.5\right| \in (0.25, 0.5)\right] + 2 \prod_{d=1}^{D} \mathbb{I}\left[\left|\frac{s_d}{H-1} - 0.5\right| \in (0.3, 0.4)\right],$$

where $H = 16$, $D = 3$ and $R_0$ is $1e-3$ in our settings.

The forward policy is implemented with the feed-forward neural network that consists of two layers with 256 hidden dimensions and is trained with a learning rate of $1e-3$. The learning rate for $Z_\theta$ is 0.1. In this task, the training is on-policy. The GFlowNets are trained with the 64 trajectories sampled from the current policy in each round. We train the pessimistic backward policy to minimize $\ell_{\text{PBP}}$ for these 64 trajectories in each round.

**Bag generation.** The implementations follow the prior study by Shen et al. [13]. The action includes one of seven types of entities in the current bag with a maximum capacity of 15. If it contains seven or more repeats of any items, it has a reward 10 with $75\%$ chance, and 30 otherwise. The threshold for determining the mode is 30.

The forward policy is implemented with the feed-forward neural network that consists of two layers with 16 hidden dimensions and is trained with a learning rate of $1e-4$. The learning rate for $Z_\theta$ is 1e-2. In this task, the training is off-policy and consists of online and offline rounds. The online round uses 32 trajectories sampled from the forward policy with an exploration rate 0.1 and the offline round uses 32 trajectories sampled from the backward policy conditioned on the high-reward objects [13]. We store these trajectories (sampled during online and offline rounds) into the buffer $\mathcal{B}$. In each round, we train the pessimistic backward policy to minimize $\ell_{\text{PBP}}$ with trajectories sampled from the buffer $\mathcal{B}$ where the $N$ in Algorithm 1 is eight. The buffer stores trajectories sampled during the previous 20 rounds.

**Maximum independent set.** The implementations follow the prior study by Zhang et al. [5]. In this task, the action is selecting a node to construct the maximum independent set. The reward is the set size and the temperature. The number of training graphs and validation graphs are 4000 and 500, respectively. The graph contains around 200 to 300 nodes. Furthermore, the reward is re-scaled with the temperature which is annealed during training, starting from 1 and ending at 500.

The forward policy is implemented with the graph isomorphism neural network [37] that consists of five layers with 256 hidden dimensions and is trained with a learning rate of $1e-3$. In this task, the training is on-policy transition-based training [5]. In each training step, the GFlowNets are trained with the 64 transitions $s \rightarrow s'$ in trajectories sampled from the current policy. We train the pessimistic backward policy to minimize the negative log-likelihood $-\log P_B(s|s')$ for these transitions within each training step.

**Molecule generation.** The overall settings is similar to the prior study [1], and our implementations are built upon the released codes. [4][5] This task aims to generate a molecule, where the action is adding a fragment. The number of available fragments is 72. The reward is computed by a pre-trained function [1]. The reward is scaled to have the maximum value nearing 1.0, and the reward exponent is set to 64.0. The mode is defined as a molecule with a reward higher than 0.97 and a Tanimoto similarity lower than 0.65, measured against previously accepted modes.

The forward policy is implemented with the graph attention transformer that consists of four layers with 128 hidden dimensions and two attention heads, which is trained with a learning rate of $1e-4$.

---

[4]https://github.com/recursionpharma/gflownet
[5]MIT License, Copyright (c) 2020 Recursion Pharmaceuticals

The learning rate for $Z_\theta$ is 1e-3. In this task, the training is off-policy, where 64 trajectories are sampled from the lagged forward policy $P_{F'}$ whose parameters are updated as $F' = 0.95F' + 0.05F$ in each round. We store these trajectories into the buffer $\mathcal{B}$. In each round, we train the pessimistic backward policy to minimize $\ell_{\text{PBP}}$ with trajectories sampled from the buffer $\mathcal{B}$ where the $N$ in Algorithm 1 is eight. The buffer stores trajectories sampled during the previous 20 rounds.

**TFBind8.** The implementations follow the prior study by Shen et al. [13]. The action appending or prepending an amino acid to the sequence with a maximum length of eight. The number of amino acids is four. The reward is pre-computed based on wet-lab measured DNA binding activity to Sine Oculis Homeobox Homolog 6 [34], which is scaled between 0.001 to 1.0. The reward exponent is set to 3.0. The mode is determined based on whether it is included in a predefined set of promising RNA sequences [13].

The forward policy is implemented with the feed-forward neural network that consists of two layers with 128 hidden dimensions and is trained with a learning rate of $1\text{e}{-4}$. The learning rate for $Z_\theta$ is 1e-2. In this task, the training is off-policy and consists of online and offline rounds. The online round uses 16 trajectories sampled from the forward policy with an exploration rate 0.01 and the offline round uses 16 trajectories sampled from the backward policy conditioned on the high-reward objects [13]. We store these trajectories (sampled during online and offline rounds) into the buffer $\mathcal{B}$. In each round, we train the pessimistic backward policy to minimize $\ell_{\text{PBP}}$ with trajectories sampled from the buffer $\mathcal{B}$ where the $N$ in Algorithm 1 is eight. The buffer stores trajectories sampled during the previous 20 rounds.

**RNA-Binding.** The implementations follow the prior study by Kim et al. [14]. The action appending or prepending an amino acid to the sequence with a maximum length of 15. The reward is scaled between 0.001 to 1.0, and the reward exponent is set to 8.0. The mode is determined based on whether it is included in a predefined set of promising RNA sequences [21].

The forward policy is implemented with the feed-forward neural network that consists of two layers with 128 hidden dimensions and is trained with a learning rate of $1\text{e}{-4}$. The learning rate for $Z_\theta$ is 1e-2. In this task, the training is off-policy and consists of online and offline rounds. The online round uses 32 trajectories sampled from the forward policy with an exploration rate 0.01 and the offline round uses 32 trajectories sampled from the backward policy conditioned on the high-reward objects [13]. We store these trajectories (sampled during online and offline rounds) into the buffer $\mathcal{B}$. In each round, we train the pessimistic backward policy to minimize $\ell_{\text{PBP}}$ with trajectories sampled from the buffer $\mathcal{B}$ where the $N$ in Algorithm 1 is eight. The buffer stores trajectories sampled during the previous 20 rounds.

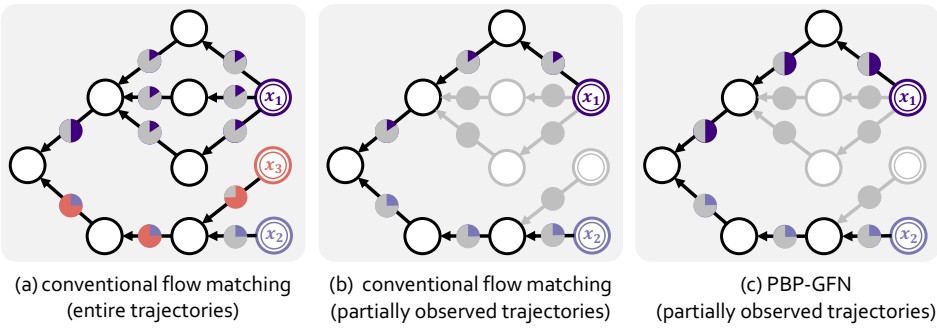

| (a) conventional flow matching | (b) conventional flow matching | (c) PBP-GFN |
| (entire trajectories) | (partially observed trajectories) | (partially observed trajectories) |

Figure 12: **A synthetic example for the case where PBP-GFN may reduce exploration.** The portion of the circle indicates the flow amount, and the color indicates the flow inducing the same-colored reward. The black and gray indicate the observed and unobserved trajectories, respectively. (**a**) The true reward of $x_1$, $x_2$, and $x_3$ are 0.5, 0.25, and 0.75, respectively. (**b-c**) Compared to the conventional GFN, PBP-GFN may yield a relatively low probability to the unobserved high-reward trajectory, by assigning a relatively high flow to the observed high-reward trajectory.

## C  Exploration-exploitation trade-off

As the case where our PBP-GFN may reduce exploration, we consider a scenario where an unobserved high-reward trajectory may largely overlap with an observed low-reward trajectory. Then, to explore the high-reward trajectory, one should assign a relatively high probability to the low-reward trajectory, i.e., the opposite of the case requiring exploitation. We exemplify and analyze this scenario in Figure 12, which illustrates how pessimistic training may reduce exploration by enhancing the exploitation of observed high-reward trajectories.

Despite the potential reduction of exploration, we would like to clarify that our method is still effective as exploitation is significant in most environments. There is no free lunch in the exploitation-exploration trade-off. One can further control the trade-off by interpolating PBP-GFN with explorative GFNs, e.g., MaxEnt, which can be an interesting future work direction.

## D   Broader impact

Improving the performance of GFLowNets can significantly influence various domains, particularly in biology (e.g., molecule and RNA sequence generation as discussed in Section 5). However, these improvements also pose potential risks including the creation of harmful drugs and the misuse of synthesized molecules.

