# OpenReview forum: "Pessimistic Backward Policy for GFlowNets"
_NeurIPS.cc/2024/Conference — NeurIPS 2024 poster_

### Official Review · Reviewer_WPBt · 2024-06-17

**Soundness:** 2
**Presentation:** 3
**Contribution:** 2
**Rating:** 6
**Confidence:** 3

**Summary:**

The paper successfully identifies and addresses the under-exploitation problem in the flow matching of GFlowNets. The proposed method, Pessimistic Backward Policy (PBP-GFN), adjusts the probabilities of backward trajectories to improve exploration and exploitation, achieving superior performance in various benchmarks.

**Strengths:**

- This paper first identifies the significant issue in conventional GFlowNets—under-exploitation due to unobserved flow in backward trajectories.
- The paper provides a clear and detailed explanation of the methodology, including the training process and the loss function used for the backward policy.

**Weaknesses:**

- The PBP-GFN introduces additional complexity to the training process of GFlowNets. Pessimistic training of backward policy might be computationally expensive or hard to tune in different settings. The author should discuss this in the updated version of the paper.
- In the experiment sections, this paper does not compare with some new GFlowNets methods, such as [1, 2], that claim improvement over the baselines. I wonder (1) Are PBP-GFN's results better than [1, 2]? (2) Can we combine the pessimistic backward policy with strategies, such as those in [1, 2], to further improve performance? I understand that each paper deals with GFlowNets from a different perspective, therefore, I am ok if not all answers are yes.
- There are some missing references that are also quite relevant to the topic. [3,4]
- I would raise my scores if my concerns are resolved.

[1] Kim, M., Yun, T., Bengio, E., Zhang, D., Bengio, Y., Ahn, S., & Park, J. (2023). Local search gflownets. arXiv preprint arXiv:2310.02710.

[2] Jang, H., Kim, M., & Ahn, S. (2023). Learning Energy Decompositions for Partial Inference of GFlowNets. arXiv preprint arXiv:2310.03301.

[3] Chen, Y., & Mauch, L. (2023). Order-Preserving GFlowNets. arXiv preprint arXiv:2310.00386.

[4] Tiapkin, Daniil, et al. "Generative flow networks as entropy-regularized rl." International Conference on Artificial Intelligence and Statistics. PMLR, 2024.

**Questions:**

- When minimizing the negative log-likelihood in Eq.5, how to ensure, or do we need to ensure that the sum of the backward probability on a state is 1, and why the total amount of backward flows will not change claimed in line 157? In the provided codebase "PBP.train_proxy", it seems the sum of log_pb_actions will change, and the total number of flows will change.

- This paper includes MIS from [1]. Can you include more difficult graph combinatorial optimization problems from [1], such as maximum cut? (Maximum clique is not necessary since it is related to MIS)

[1] Zhang, D., Dai, H., Malkin, N., Courville, A., Bengio, Y., & Pan, L. (2023). Let the flows tell: Solving graph combinatorial optimization problems with gflownets. arXiv preprint arXiv:2305.17010.

**Limitations:**

Please see the **Weaknesses** part.

---

> ### Author Rebuttal · Authors · 2024-08-07
>
> Dear reviewer WPBt,
>
> We express our deep appreciation for your time and insightful comments. In what follows, we address your comments one by one.
>
> ---
>
> **W1. The PBP-GFN introduces additional complexity, which may be expensive or inpractical in some settings. Can the authors provide a discussion on this point?**
>
> We clarify that the training of the pessimistic backward policy does not require significant overhead and is practical in most cases, as it only requires computing the gradients of the negative log-likelihood for the given trajectories. This requires similar or higher complexity than the conventional backward policy but less than the sub-structure backward policy. Especially, this complexity (i.e., the computation of gradients) would be minor for real-world problems with expensive reward evaluations, e.g., molecular docking tools or wet-lab experiments. We will update our future manuscript to reflect these points.
>
> ---
>
> **W2. Compared to independent lines of research for exploitation, e.g., local search GFN and LED-GFN, (A) does PBP-GFN show superior results, and (B) can we combine PBP-GFN with these approaches to further improve performance?**
>
> To address your comments, we conducted new experiments by (A) comparing PBP-GFN with local search and LED-GFN and (B) combining PBP-GFN with them. Note that (B) is trivial since our pessimistic backward policy is orthogonal to local search and LED-GFN that address off-policy sampling and local credits.
>
> We present the results in **Figure C(a)** and **Figure D(a)** of our rebuttal PDF. One can see that PBP-GFN (A) shows similar performance compared to local search GFN and LED-GFN, and (B) further improves performance when combined with them.
>
> ---
>
> **W3. There are some missing references that are also quite relevant to the topic [2,3].**
>
> Thank you for your valuable suggestions! We will incorporate them into the related works section of our future manuscript.
>
> ---
>
> **Q1. (A) How to ensure the sum of the backward probability on a state is $1$, and (B) the provided codebase "PBP.train_proxy" seems change the sum of `log_pb_actions`, so the total backward flows will change.**
>
> We would like to clarify that (A) is trivial. When we implement the backward policy using softmax, i.e., $\sum_{s\in\text{parent}(s')}P_B(s|s')=1$, the summation of the backward probability on a state satisfies $\sum_{\tau\in\mathcal{T}(x)}P_B(\tau|x)=\sum_{(s_0\rightarrow\cdots\rightarrow s_T)\in\mathcal{T}(x)} {\prod_{t=0}^{T-1}P_B(s_t |s_{t+1})=1}$, by iterating the law of total probability over entire trajectories. Note that we do not constrain $\sum_{\tau\in\mathcal{B}(x)}P_B(\tau|x)$ for observed trjajectories $\tau \in \mathcal{B}(x)$ to be equal to one, but maximize it.
>
> For (B), we would like to clarify that the sum of `log_pb_actions` corresponds $\log P_B(\tau|x)=\sum_{t=0}^{T-1} \log P_B(s_t|s_{t+1})$ for the given trajectory. The change of this does not affect the total amount of backward flow, i.e, $R(x)\sum_{\tau \in \mathcal{T}(x)}P_B(\tau|x)$, as $\sum_{\tau \in \mathcal{T}(x)}P_B(\tau|x)=1$ is always guaranteed over the entire trajectories.
>
>
> ---
>
> **Q2. This paper includes MIS. Can you include more difficult graph combinatorial optimization problems from [1], such as maximum cut?**
>
> Indeed, we previously tried other graph combinatorial optimization problems from [1], e.g., the maximum cut problem, but could not reproduce the official results even when using the official implementation. For example, the official score for the maximum cut problem is around $700$, but running their implementation yields a score of $2000$ during the initial training round. To alleviate your concerns, we report the result of PBP-GFN applied to the maximum cut problem in **Figure D(c)** of our rebuttal PDF.
>
> ---
>
> [1] Let the Flows Tell: Solving Graph Combinatorial Problems with GFlowNets, NeurIPS 2023
>
> [2] Order-Preserving GFlowNets, ICLR 2024
>
> [3] Generative Flow Networks as Entropy-Regularized RL, AISTATS 2024

---

> > ### Comment · Reviewer_WPBt · 2024-08-12
> >
> > Thank you for the detailed clarifications and additional experiments. I decide to raise my score to 6.

---

> ### Author Response · Authors · 2024-08-12
>
> Dear reviewer WPBt,
>
> We are happy to hear that our efforts have addressed your concerns! We also appreciate your insightful comments on our works.

---

### Official Review · Reviewer_CTvM · 2024-06-30

**Soundness:** 3
**Presentation:** 3
**Contribution:** 2
**Rating:** 5
**Confidence:** 4

**Summary:**

The authors present a problem with GFlowNets training that originates from not having seen backward trajectories for a particular terminal state. The authors show that because of the lack of observed trajectories, the backward flows underestimate the probabilities for the observed flows, resulting in a (forward) policy that may not match the reward distribution. The authors propose to mitigate the problem by increasing backward flows for the observed trajectories. The authors provide experimental results in eight different environments, showing the superior performances of their method in terms of mode discovery and distribution fitting.

**Strengths:**

1. Well-written motivation for the problem described.
2. The solution presented is well-motivated and clearly described.
3. The experiments include well-studied environments and baselines, showing the method's performance in exploitation.

**Weaknesses:**

1. The discussion about lacking backward flow given the unseen trajectories is valid. In practice, one can mitigate this by having a uniform backward policy with **a reward-prioritized replay buffer** [1], which also can increase the probability a trajectory yielding high rewards, thereby tackling the mentioned problem. I feel adding this baseline will improve the paper.

2. As an exploitation method, at least one experiment with a larger state-space should be useful to show its performance in larger state-spaces. Hypergrid with a larger dimension and horizon can be one option for this.


[1] Shen, Max W., et al. "Towards understanding and improving gflownet training." International Conference on Machine Learning. PMLR, 2023.

**Questions:**

1. What is the hypermeters for GAFN in Figure 9? Were the hyperparameters tuned for the experiments?
2. The example in Figure 3 raises a question: if $P_B(x|\tau)[i]$ is maximized, $P_B(x|\tau)[j] \approx 0; j \neq i$, thereby matching it will cause $P_F(x|\tau)[j] \approx 0; j \neq i$. In smaller state-space, it may not be a problem, but in a larger state-space, this should be an issue as it may cause $p(x) \approx 0$ where x necessitates action j whose probability we've just made approximately 0. Curious to hear what you think about this/whether you've done experiments in large-scale environments.

**Limitations:**

Yes. Limitations are adequately addressed.

---

> ### Author Rebuttal · Authors · 2024-08-07
>
> Dear reviewer CTvM,
>
> We express our deep appreciation for your time and insightful comments. In what follows, we address your comments one by one.
>
> ---
>
> **W1. One can mitigate under-exploitation using a reward-prioritized replay buffer. Adding this as a baseline will be helpful.**
>
> We clarify that our experiments on the bag and RNA sequence have already incorporated the reward-prioritized buffer into all baselines and our method (**Appendix B**). Since the pessimistic backward policy and the reward-prioritized buffer are orthogonal, they are not directly comparable and one can combine them to train GFlowNets.
>
> Nevertheless, to further address your comments, we also considered PBP-GFN without a reward-prioritized buffer and compared it with baselines using a reward-prioritized buffer. We present results in **Figure C(b)** of our rebuttal PDF. One can observe that the pessimistic backward policy (1) yields similar performance improvements compared to the reward-prioritized buffer, and (2) further improves their performance when combined with it.
>
> ---
>
> **W2. As an exploitation method, at least one experiment with a larger state-space could be useful, e.g., larger hyper-grid.**
>
> We first clarify that our molecule generation environment considers a sufficiently large state space sized up to $10^{16}$ [1]. To further address your suggestion about a larger hyper-grid, we conducted additional experiments on $40\times 40 \times 40 \times 40$ hyper-grid. To the best of our knowledge, this is larger than any hyper-grid experiment conducted in the literature. We present the results in **Figure D(b)** of our rebuttal PDF. One can observe that our method still shows superior results compared to considered baselines.
>
> ---
>
> **Q1. What are the hyperparameters for GAFN?**
>
> For GAFN, we searched for the coefficient $\alpha$ of intrinsic rewards within $\{1 \mathrm{e-}1, 1 \mathrm{e-}2, 1 \mathrm{e-}3\}$. We designed the random network to have the same architecture as the policy network, but with an output dimension of one, and used a learning rate of $1 \mathrm{e-}3$ for this network.
>
> ---
>
> **Q2. In a large state space, PBP-GFN may cause $P_F(x)\approx 0$ when $x$ necessitates an action $j$ but PBP-GFN have made probability of $j$ approximately $0$. Curious to hear what you think about this and whether you've done experiments in large-scale environments.**
>
> We think that such a potential reduction in the exploration of unobserved objects is natural due to the exploration-exploitation trade-off. As PBP-GFN enhances the exploitation, this may reduce exploration for the unobserved objects in the large state space (discussed in **Limitation** section). However, we would like to clarify that one can simply relax this issue by reducing the learning rate for the pessimistic backward policy or incorporating an exploration-focused off-policy sampling method to observe $x$ with $P_F(x)\approx 0$.
>
> Furthermore, it is worth noting that our method has already shown good performance in a large state space, e.g., molecule generation sized up to $10^{16}$. This implies that our method is still effective in environments with large state space, as exploitation is also significant for them.
>
> ---
>
> [1] Trajectory balance: Improved credit assignment in GFlowNets, NeurIPS 2022

---

> > ### Comment · Reviewer_CTvM · 2024-08-12
> >
> > Thank you for your rebuttal. I think the method’s promise is weakened because of Q2. Since it is an exploration method, further analysis of this phenomenon in a large state space is required. Hence, I would like to keep my score.

---

> ### Author Response · Authors · 2024-08-12
>
> Dear reviewer CTvM,
>
> Thanks you for your response. We appreciate your insightful and constructive feedback for improving our paper in many aspects.
>
> Additionally, we would like to emphasize that we have already empirically analyzed the reduction of exploration (as an exploitation method) in large state spaces by measuring the diversity of high-score objects (**Figure 7(c)**). However, we observed no particular reduction in practice. In our future manuscript, we will also incorporate an analysis of the case where our PBP-GFN may reduce exploration (**Figure B** of rebuttal PDF) and discuss mitigation strategies with contents from (**u41F-W4,5**). We hope that this addresses your concerns.

---

### Official Review · Reviewer_smvU · 2024-07-13

**Soundness:** 2
**Presentation:** 3
**Contribution:** 2
**Rating:** 5
**Confidence:** 3

**Summary:**

This paper proposes a pessimistic backward policy for GFlowNets (GFN), which maximizes the expectation of observed backward trajectories. This paper points out the under-exploitation of high-reward objects for previous GFN training methods and provides a deep analysis of this problem. Extensive experiments validate the superior performance of the proposed method.

**Strengths:**

- This paper is well-written. The illustrations of the proposed method are clear, and the toy examples provided are easy to understand.
- This paper proposes a simple yet effective method to address the problems of under-exploitation in GFN training.
- Extensive experiments on 8 benchmarks validate the efficacy of the method.

**Weaknesses:**

Does the auxiliary objective introduced in Eq. 5 affect the original GFN training objective? Since higher probabilities are assigned to the backward transitions of observed trajectories, could this impact the original assumptions of TB, DB, and other objectives? Do you observe instability during training after some episodes, once the model has seen a sufficient number of samples?

**Questions:**

- How do you estimate Eq. 5 in practice? This is an important component of the method, but it is not described.
- What's the meaning of $N$ in algorithm 1? Why does this method need multi-round gradient updates?

**Limitations:**

Limitations have been discussed in this paper.

---

> ### Author Rebuttal · Authors · 2024-08-07
>
> Dear reviewer smvU,
>
> We express our deep appreciation for your time and insightful comments. In what follows, we address your comments one by one.
>
> ---
>
> **W1. Does the pessimistic training objective affect the original assumption of GFlowNets training objective?**
>
>
> Our pessimistic training objective does not affect the original assumption of GFlowNet objectives, e.g., DB, TB, and subTB. The assumption of them, i.e., flow matching over entire trajectories $\tau \in \mathcal{T}$ constructs target Boltzmann distribution, is valid for any design of backward policy, i.e., degree of freedom [1,2]. Our pessimistic training objective only modifies this backward policy within the GFlowNet objectives.
>
> ---
>
> **W2. Did you observe any instability during training after the model had seen a sufficient number of samples?**
>
> In our experimental setup, we observed no particular instability after the model had seen a sufficient number of samples.
>
> ---
>
> **Q1. How do you estimate Equation (5) in practice?**
>
> In practice, we estimate the gradients of **Equation (5)** using the stochastic gradients computed with mini-batch randomly sampled from the buffer $\mathcal{B}$, as described in **Algorithm 1**. We will update our future manuscript to better reflect this point.
>
> ---
>
> **Q2. What's the meaning of $N$ in Algorithm 1, and why does pessimistic training require multi-round gradient updates with $N$?**
>
> As you mentioned, $N$ is a hyperparameter that specifies the number of update rounds for training the pessimistic backward policy. We require this as we use stochastic gradients to minimize **Equation (5)**, which involves multi-round updates over multiple mini-batches.
>
> ---
>
> [1] GFlowNet Foundations, JMLR 24
>
> [2] Trajectory balance: Improved credit assignment in GFlowNets, NeurIPS 2022

---

> > ### Comment · Reviewer_smvU · 2024-08-10
> >
> > Thanks for your response and hard work. After reading your rebuttal and other reviews, I've decided to maintain my score.

---

> ### Author Response · Authors · 2024-08-12
>
> Dear reviewer smvU,
>
> We also appreciate your insightful comments. Thank you again for your time and effort for improving our paper!

---

### Official Review · Reviewer_u41F · 2024-07-13

**Soundness:** 3
**Presentation:** 3
**Contribution:** 3
**Rating:** 3
**Confidence:** 4

**Summary:**

This paper addresses the under-exploitation of high-reward objects in Generative Flow Networks (GFlowNets) due to the limited observation of trajectories during training. The authors propose PBP-GFN (Pessimistic Backward Policy for GFlowNets), which modifies the backward policy to maximize the observed backward flow, aligning it closer to the true reward. They argue that this pessimistic training scheme encourages the forward policy to assign higher probabilities to high-reward objects, even if they are under-represented in the observed trajectories. Experiments across eight benchmarks, including hypergrid, molecule, and RNA sequence generation, demonstrate that PBP-GFN improves the discovery of high-reward objects while maintaining object diversity.

**Strengths:**

* The paper tackles an important issue in GFlowNet training – the potential for under-exploration of high-reward objects due to the sparsity of observed trajectories.  This issue is particularly relevant in complex domains with vast trajectory spaces.
* The proposed solution, PBP-GFN, is conceptually simple and intuitive. Maximizing the observed backward flow to align with the true reward directly addresses the identified problem of under-exploitation.
* The paper provides extensive experimental results across a variety of tasks, showcasing the effectiveness of PBP-GFN in enhancing the discovery of high-reward objects.

**Weaknesses:**

* While the intuition behind PBP-GFN is clear, the paper lacks a strong theoretical analysis to support its claims. The "error bound" analysis in Appendix A is limited in scope and doesn't offer a comprehensive understanding of the algorithm's convergence properties or its impact on the target Boltzmann distribution.
* The core idea of modifying the backward policy in GFlowNets has been explored in prior works [1, 2, 15]. PBP-GFN, while achieving promising empirical results, doesn't present a significant conceptual departure from these existing approaches.  Its primary contribution appears to be a specific strategy for maximizing the observed backward flow, but the paper lacks a detailed comparison and analysis of how this strategy differs from or improves upon existing techniques.

* The paper primarily focuses on metrics like the number of modes discovered and the average top-100 score. While these metrics are relevant for measuring exploration and exploitation, they provide a limited view of the overall quality and diversity of generated samples.  Evaluating PBP-GFN on more comprehensive downstream task-specific metrics would strengthen the paper's claims.

*  While the paper claims that PBP-GFN maintains object diversity, the pessimistic training scheme inherently introduces a bias towards the observed trajectories. This bias could potentially lead to a reduction in exploration and the generation of degenerate or less diverse samples, especially in domains with significant uncertainty or where observed trajectories are not truly representative of the target distribution.  A deeper analysis and empirical evaluation of this potential bias would be useful.

* The performance of PBP-GFN heavily relies on the quality and representativeness of the observed trajectories.  In scenarios where the initial observed trajectories are biased or incomplete, PBP-GFN could amplify these biases, hindering the discovery of truly novel and high-reward objects. The paper doesn't address this sensitivity or propose mitigation strategies.

* The paper mainly compares PBP-GFN with other backward policy design methods. However, it doesn't provide a thorough comparison with other exploitation-focused techniques in GFlowNets, such as local search GFlowNets [3] or those focusing on higher-reward trajectories [2]. This limited comparison weakens the paper's claim of achieving superior performance in discovering high-reward objects.

[1] Trajectory balance: Improved credit assignment in gflownets
[2] Towards understanding and improving gflownet training.
[3] Local search gflownets
[4] Maximum entropy gflownets with soft q-learning

**Questions:**

Please refer to weaknesses section above.

**Limitations:**

N/A.

---

> ### Author Rebuttal · Authors · 2024-08-07
>
> Dear reviewer u41F,
>
> We express our deep appreciation for your time and insightful comments. In what follows, we address your comments one by one.
>
> ---
>
> **W1. The paper lacks a strong theoretical analysis to support claims.**
>
> We agree that our paper lacks a strong theoretical analysis since we mainly focus on empirical performance improvements. We note that a strong theoretical analysis would be challenging (though valuable) since it requires thinking about the optimization landscape of GFlowNets, where no theoretical results have been made. Hence, we chose to provide insights in Appendix A by analyzing our PBP-GFN under strong assumptions.
>
> ---
>
> **W2. Compared to existing backward policies [1,2,3], this work (A) does not present a significant conceptual departure and (B) lacks a detailed comparison.**
>
> We respectfully disagree with (A). Unlike existing backward policies that focus on under-exploration or credit assignment issues, PBP-GFN is designed to tackle a new under-exploitation problem stemming from the unobserved backward flow.
>
> Next, to resolve (B), we will incorporate the following detailed comparisons with the existing backward policies in our future manuscript.
>
> - While **uniform backward policy [1]** and **MaxEnt backward policy [3]** assign a fixed probability to the backward transition for enhancing exploration, our pessimistic backward policy learns the probability of the backward transition for enhancing exploitation.
> - While **conventional backward policy [1]** and **sub-structure backward policy [2]** may enhance the exploitation by learning the backward flow to align with the forward flow or to improve the credit assignments, they do not directly reduce the unobserved backward flow and do not resolve the under-exploitation stemming from that.
>
> ---
>
> **W3. Current metrics are insufficient to capture overall quality and diversity. Downstream task-specific metrics would strengthen the paper's claims.**
>
> We would like to clarify that we have already incorporated downstream task-specific metrics following prior studies [1,3,4]. We consider the Tanimoto similarity and edit distance to measure the diversity of molecules and RNA sequences, respectively.
>
> To further address your comments on the overall sample quality, we provide the relative mean error [2] which measures the distance between the mean values of the empirical generative distribution and the target Boltzmann distribution. We present the results in **Figure A** of our rebuttal PDF. One can see that our method shows superior performance.
>
> ---
>
> **W4. PBP-GFN may reduce exploration in domains with significant uncertainty or where trajectories are not representative of the target distribution. A deeper analysis and empirical evaluation of this are recommended.**
>
> To address your comments, we consider a scenario where the observed trajectories are not representative of the target distribution. In this domain, an unobserved high-reward trajectory may largely overlap with an observed low-reward trajectory. Then, to explore the high-reward trajectory, one should assign a relatively high probability to the low-reward trajectory, i.e., the opposite of the case requiring exploitation.
>
> We examplify and analyze this scenario in **Figure B** of our rebuttal PDF, which (1) illustrates how pessimistic training may reduce exploration by enhancing the exploitation of observed high-reward trajectories and (2) provides an empirical evaluation of the exploration. One can observe that PBP-GFN may reduce the exploration in considered settings.
>
> Despite the potential reduction of exploration, we would like to clarify that our method is still effective as exploitation is significant in most environments. As discussed in **Limitation** section, there is no free lunch in the exploitation-exploration trade-off. One can futher control the trade-off by interpolating PBP-GFN with explorative GFNs, e.g., MaxEnt, which can be an interesting future work direction.
>
> ---
>
> **W5. PBP-GFN can amplify biases in initially observed trajectories, hindering the discovery of novel objects, but the paper does not address this or mitigation strategies.**
>
> As you mentioned, our method may struggle to discover novel objects when heavily biased towards initially observed trajectories. However, we would like to clarify that such failure cases are rare in practice, as demonstrated in extensive experiments.
>
> Furthermore, to mitigate the bias towards initially observed trajectories, one can reduce the learning rate of the pessimistic backward policy in the initial training rounds. Additionally, incorporating exploration-focused sampling, e.g., a noisy policy, can address this issue. We will add a discussion about this potential risk and mitigation strategies in our future manuscript.
>
> ---
>
> **W6. The paper lacks a comparison with other exploitation techniques, e.g., local search or reward-prioritized buffer.**
>
> First, we clarify that local search and reward-prioritized buffers are sampling methods for off-policy training, that are orthogonal to our pessimistic backward policy, i.e., the methods are not directly comparable. One can combine both approaches to improve performance further. Note that we have already used the reward-prioritized buffer to implement our method and baselines (**Appendix B**).
>
> Nevertheless, to address your comments, we verify the effectiveness of PBP-GFN by comparing or combining the pessimistic backward policy with local search and reward-prioritized buffer. We present the results in **Figure C** of our rebuttal PDF. One can observe that our method (1) shows similar performance compared to other techniques and (2) consistently improves the performance when combined with them.
>
> ---
>
> [1] Trajectory balance: Improved credit assignment in GFlowNets, NeurIPS 2022
>
> [2] Towards understanding and improving GFlowNet training, ICML 2023
>
> [3] Maximum entropy GFlowNets with soft Q-learning, AISTATS 2024
>
> [4] Local search GFlowNets, ICLR 2024

---

> > ### Author Response · Authors · 2024-08-14
> >
> > Dear Reviewer u41F,
> >
> > We appreciate your constructive feedback for improving our paper in many aspects. We have provided the requested analysis, discussions, and experiments. We are curious whether our rebuttal has resolved your concerns. Thank you again for your time and effort.

---

### Author Rebuttal · Authors · 2024-08-07

Dear reviewers (**u41F**, **smvU**, **CTvM**, and **WPBt**) and area chairs,

We are deeply grateful for the time and effort you spent reviewing our manuscript. In what follows, we summarize our rebuttal PDF and planned revisions.

---

### Summary of rebuttal PDF

Our rebuttal PDF provides the following contents:

- **Figure A** provides experiments on a new metric for the overall quality of generated samples (**u41F-W3**).
- **Figure B** presents a synthetic example for the case where our PBP-GFN may reduce exploration (**u41F-W4**).
- **Figure C** and **Figure D(a)** present the results of comparing and combining PBP-GFN with methods that improve exploitation (**u41F-W6**, **CTvM-W1**, **WPBt-W2**).
- **Figure D(b)** and **Figure D(c)** present the results on additional benchmarks (**CTvM-W2**, **WPBt-Q2**).

---

### Summary of planned revisions

We will update our future manuscripts to incorporate the following contents that have been addressed in our rebuttal:

- Clarification of the detailed conceptual differences with existing methods for backward policy modification (**u41F-W2**)
- Clarification of detailed implementations for pessimistic training (**smvU-Q1**)
- Additional references relevant to the topic (**WPBt-W3**)
- Additional experimental results measuring overall sample quality (**u41F-W3**)
- Additional experimental results comparing with other methods that improve exploitation (**u41F-W6**, **CTvM-W1**, **WPBt-W2**)
- Additional experimental results on the large-scale hyper-grid (**CTvM-W2**)
- Analysis for the case where our PBP-GFN may reduce exploration (**u41F-W4**)
- Discussion of the strategies for mitigating risks in exploitation (**u41F-W4,5**)
- Discussion of additional time complexity (**WPBt-W1**)

---

### Decision · Program_Chairs · 2024-09-25

**Decision:**

Accept (poster)

**Comment:**

This paper explores the under-exploitation problem in the flow matching of GFlowNets and proposes PBP-GFN.

There were some concerns about the lack of theoretical analysis, which should not be a reason for rejection. The authors addressed most of the concerns raised by the reviewers in their rebuttal. Most importantly, the authors compared their method with a couple of baselines suggested by the reviewers. Even though the proposed method achieves comparable performance to these baselines, they are orthogonal, and the authors show that one can gain more by combining PBP-GFN with these baselines. I recommend an acceptance.